# Tandem Reactions Based on the Cyclization of Carbon Dioxide and Propargylic Alcohols: Derivative Applications of α-Alkylidene Carbonates

Bowen Jiang [1,2,†], Xiangyu Yan [1,2,†], Yong Xu [1,2], Natalya Likhanova [3], Heriberto Díaz Velázquez [3], Yanyan Gong [4], Ye Yuan [1,*] and Francis Verpoort [1,2,5,*]

1   State Key Laboratory of Advanced Technology for Materials Synthesis and Processing, Wuhan University of Technology, Wuhan 430070, China; swedenjiang@whut.edu.cn (B.J.); 303600@whut.edu.cn (X.Y.); 303906@whut.edu.cn (Y.X.)

2   School of Materials Science and Engineering, Wuhan University of Technology, Wuhan 430070, China

3   Dirección de Investigación en Transformación de Hidrocarburos, Instituto Mexicano del Petróleo, Eje Central Lázaro Cárdenas 152, San Bartolo Atepehuacan, Mexico City 07730, Mexico; nvictoro@imp.mx (N.L.); hdiaz@imp.mx (H.D.V.)

4   State Key Laboratory of Biobased Material and Green Papermaking, Qilu University of Technology, Shandong Academy of Sciences, Jinan 250353, China; kxf@whut.edu.cn

5   National Research Tomsk Polytechnic University, Lenin Avenue 30, 634050 Tomsk, Russia

\*   Correspondence: fyyuanye@whut.edu.cn (Y.Y.); francis@whut.edu.cn (F.V.)

†   These authors contributed equally to this work.

**Abstract:** As a well-known greenhouse gas, carbon dioxide ($CO_2$) has attracted increasing levels of attention in areas of energy, environment, climate, etc. Notably, $CO_2$ is an abundant, nonflammable, and renewable C1 feedstock in view of chemistry. Therefore, the transformation of $CO_2$ into organic compounds is an extremely attractive research topic in modern green and sustainable chemistry. Among the numerous $CO_2$ utilization methods, carboxylative cycloaddition of $CO_2$ into propargylic alcohols is an ideal route due to the corresponding products, α-alkylidene cyclic carbonates, which are a series of highly functionalized compounds that supply numerous potential methods for the construction of various synthetically and biologically valuable agents. This cyclization reaction has been intensively studied and systematically summarized, in the past years. Therefore, attention has been gradually transferred to produce more derivative compounds. Herein, the tandem reactions of this cyclization with hydration, amination, alcoholysis, and isomerization to synthesize α-hydroxyl ketones, oxazolidinones, carbamates, unsymmetrical carbonates, tetronic acids, ethylene carbonates, etc. were systematically reviewed.

**Keywords:** carbon dioxide transformation; tandem reactions; propargylic alcohols; α-alkylidene carbonates; green synthesis

## 1. Introduction

Since the industrial revolution, fossil fuels have become the primary energy source for human beings. Simultaneously, continuous consumption of fossil fuels has produced excessive carbon dioxide ($CO_2$), which leads to global warming and causes severe environmental impact such as rising sea levels, frequent extreme weather events, and imbalances of the ecosystem [1–4]. The threat of global warming to the environment and climate has been a significant challenge for human beings in the 21st century. Accordingly, the capture and treatment of $CO_2$ has become a strategic priority and has been intensively investigated [5]. In this aspect, the carbon capture and utilization (CCU) strategy has been regarded as a promising option for controlling the accumulation of $CO_2$ [6–13]. Compared with the traditional carbon capture and storage (CCS) strategy, CCU eliminates the extra energy-consuming process of $CO_2$ desorption and compression whereby $CO_2$ can be

captured and transformed directly into valuable compounds [14–16], fuels [17,18], and materials [19–23]. In other words, the CCU strategy can not only reduce the level of $CO_2$ amount but bring a series of economic and environmental benefits [24]. Consequently, CCU is an essential way to successfully achieve carbon neutrality, which meets the requirements of sustainable development.

From the standpoint of green chemistry, $CO_2$ is an abundant, nonflammable, and inexpensive carbon resource [5,25], which makes it a feasible substitute for some non-renewable and toxic chemicals such as phosgene, cyanic acid, and carbon monoxide in various synthetic processes [26,27]. Hence, the catalytic conversion of $CO_2$ has become an extremely attractive field in modern green chemistry [28–30]. However, the inherent thermodynamic stability and kinetic inertness of $CO_2$ remain the main obstacles to its effective conversion [31]. Generally, strong nucleophiles or harsh reaction conditions (high temperature or high $CO_2$ pressure) were required when $CO_2$ was utilized in CCU processes [32]. Therefore, developing new synthetic routes and high-efficiency catalysts are urgently needed in this area.

Hitherto, many strategies have been developed to capitally transform $CO_2$ into high value-added chemicals such as carbonates, aldehydes, ketones, carboxylic acids, esters, amides, alkanes, quinazolines, and so forth [33,34]. Among them, $\alpha$-alkylidene cyclic carbonates, fabricated by the cyclic carboxylation of $CO_2$ with propargylic alcohols, are a class of representative and meritorious compounds with a wide range of applications, such as polar aprotic solvents, electrolytes in batteries, monomers in the synthesis of polycarbonates, and more importantly as reaction intermediates in the manufacture of fine chemicals [27,35]. Therefore, scientists have gradually focused on the derivative tandem reactions based on these $\alpha$-alkylidene cyclic carbonates, such as their combinations with amination, hydration, alcoholysis, isomerization, etc. Specifically, multicomponent reactions are one-pot reactions adopting more than two different kinds of raw materials, which offers numerous remarkable advantages such as easy operation, facile automation, high atom economy, simple separation and purification process, and reduced generation of by-products [36]. Based on this, the synthesis of carbamates, oxazolidinones, $\alpha$-hydroxyl ketones, non-symmetric carbonates, cyclic carbonates, and even some polymers like polyurethanes and polycarbonates via three-component tandem reaction of $CO_2$ and propargylic alcohols with respective amines [37–39], $H_2O$ [40–43], alcohols [33,38,44–52], and 2-aminoethanols [53–58] in the presence of high-efficiency catalysts such as transition metals, organic bases, ionic liquids (ILs), and heterogeneous catalysts has been recently developed with explosive growth (Figure 1). Apart from the multicomponent tandem reactions, an in situ transformation of these $\alpha$-alkylidene cyclic carbonates was also developed. The tetronic acids could be smoothly derived from the isomerization of the corresponding cyclic carbonates [59]. In this review, multicomponent tandem reactions of $CO_2$, propargylic alcohols with nucleophiles by the one-pot method and their mechanisms were expounded through the systematic investigations on the reports in the past few years. In addition, the rarely reported isomerization reactions of cyclic carbonates were also briefly introduced.

**Figure 1.** Schematic diagram of several routes for $CO_2$ transformation.

## 2. Three-Component Reactions of Propargylic Alcohols, $CO_2$ and Amines

As a series of typical nucleophilic reagents, amines can be employed in the three-component tandem reactions with $CO_2$ and propargylic alcohols to effectively prepare oxazolidinones or carbamates, both of which are vital skeletons in biology, pharmaceutical chemistry and organic synthesis [60–64]. These three-component reactions undertake a two-step procedure. In the first step, the cyclic carbonate intermediates **M1** are formed by the carboxylative cyclization of $CO_2$ with the propargylic alcohols. The next step is the nucleophilic attack of the nitrogen atoms in amines to the carbonyl groups in **M1**. After this ring-opening process, the intramolecular cyclization or the keto-enol tautomerism occurs and the desired oxazolidinones **1**, oxazolones **2** or carbamates **3** are generated. Generally, the three-component reaction of tertiary propargylic alcohols, $CO_2$ and primary amines affords the 4-methyleneoxazolidin-2-ones (**1**); the three-component reaction of primary or secondary propargylic alcohols, $CO_2$ and primary amines affords 4-methyloxazol-2-ones (**2**); and the three-component reaction of propargylic alcohols, $CO_2$ and secondary amines affords β-oxopropylcarbamates (**3**), as illustrated in Scheme 1.

**Scheme 1.** The process of the three-component reactions of propargylic alcohols, $CO_2$ and amines.

### 2.1. Oxazolidinones as Products

In 1987, Sasaki and Dixneuf [65] pioneered the three-component tandem reaction of $CO_2$, propargylic alcohols, and primary amines employing the ruthenium complex as the catalyst. With the participation of $Ru_3(CO)_{12}$, 2-oxo-l,3-oxazolines were generated in 13–28% yields from the reaction of prop-2-yn-1-ol or 3-butyn-2-ol with *n*-propylamine under 5 MPa of $CO_2$ in $CH_3CN$ (Scheme 2).

**Scheme 2.** $Ru_3(CO)_{12}$ catalyzed three-component reaction of $CO_2$, propargylic alcohols and primary amines.

Until 2005, a relatively economical copper(I) catalyst was developed for this three-component reaction by Deng et al. [66], who reported a type of recyclable CuCl/1-butyl-3-methylimidazolium tetrafluoroborate ([BMIm]$BF_4$) system. With this catalyst, 4-methylene-2-oxazolidinones could be obtained in 78–95% yields from the corresponding propargylic alcohols and primary amines under 2.5 MPa of $CO_2$ (Scheme 3). Furthermore, the CuCl/[BMIm]$BF_4$ system could be reused at least 3 times without obvious activity decline. Unfortunately, when primary and secondary propargylic alcohols or aniline were applied as one of the substrates, the desired oxazolidinone products could not be formed. In 2007, Jiang et al. [67] proposed a superior copper-catalyzed process in which CuI was utilized under supercritical $CO_2$ ($scCO_2$) conditions. Both 4-methyleneoxazolidin-2-ones and 4-methyloxazol-2-ones were obtained from the reactions of various primary propargylic alcohols and secondary propargylic alcohols with primary amines in 70–95% and 88–96% yields (Scheme 4), indicating its broad substrate scope. Shortly afterwards, Zhao et al. [68] found that CuCl was also an effective catalyst without the participation of ILs, which enabled this three-component reaction under solvent-free and atmospheric $CO_2$ pressure. Although excess primary amines were required and the yields were relatively low, the oxazolidinones could be obtained through this facile and mild process (Scheme 5).

In 2019, Wang et al. [69] reported a CuI/tetrabutylphosphoniumimidazol ([P$_{4444}$][Im]) system for the transformation of $CO_2$ into oxazolidinones under room temperature and atmospheric $CO_2$ pressure (Scheme 6). The corresponding products were obtained in 54–87% yields. However, anilines also failed to generate the oxazolidinones. Regarding the mechanism, CuI and [P$_{4444}$][Im] synergistically catalyzed the reaction. CuI activated

the C≡C bonds of the propargyl alcohols; meanwhile, $[P_{4444}][Im]$ activated the hydroxyl groups of the propargyl alcohols. Furthermore, the authors proposed that the catalytic activity was affected by the alkalinity of ILs. A higher pKa would enhance its $CO_2$ fixation ability in this three-component reaction, which led to an improved reaction performance of $CO_2$, propargylic alcohols and primary amines. On the other hand, the dissociation ability of the metal anions played a pivotal role in the activity of the halogenated copper salts. Generally, a stronger dissociation ability of the anions would result in the better catalytic activity of the corresponding copper salt.

R₁ = alkyl
R₂ = alkyl, Ph
R₃ = alkyl, benzyl, allyl, Py

**Scheme 3.** $CuCl/[BMIm]BF_4$ system catalyzed three-component reaction of $CO_2$, propargylic alcohols and primary amines.

R₁ = H, alkyl, aryl
R₂ = H, alkyl, aryl
R₃ = alkyl

**Scheme 4.** CuI catalyzed three-component reaction of $CO_2$, propargylic alcohols and primary amines.

R₁ = H, Me
R₂ = H, Me, Ph, *i*-Pr, *n*-C₅H₁₃
R₁, R₂ = -(CH₂)₅-
R₃ = *n*-Bu, *s*-Bu, allyl, Cy

**Scheme 5.** CuCl catalyzed three-component reaction of $CO_2$, propargylic alcohols and primary amines.

R₁ = Me
R₂ = Me, Et, *i*-Bu
R₁, R₂ = -(CH₂)₅-
R₃ = *n*-Bu, *i*-Bu, *n*-Hex, Cy, Ph

**Scheme 6.** $CuI/[P_{4444}][Im]$ system catalyzed three-component reaction of $CO_2$, propargylic alcohols and primary amines.

Apart from the copper(I) compounds, silver salts had been found to be an excellent catalyst for this three-component reaction. In 2009, Jiang and Zhao [70] reported that AgOAc could efficiently promote the reaction of internal propargylic alcohols with primary amines in scCO$_2$. It was noteworthy that when internal propargylic alcohols were used as substrates, both the primary and secondary alcohols could participate in this reaction, providing corresponding oxazolidinones rather than oxazolones (Scheme 7). In 2014, He group [71] developed a AgWO$_4$/Ph$_3$P system that worked under solvent-free conditions and 0.5 MPa of CO$_2$. The oxazolidinones derivatives were obtained in 83–95% yields (Scheme 8). Notably, this catalytic system was not sensitive to air and moisture, which facilitated the operations of the CO$_2$ conversion process. Recently, Zhang and He et al. [72] demonstrated that the 1,3-oxazolidin-2-one derivatives could be achieved from the three-component reaction of typical terminal propargylic alcohols, CO$_2$, and diverse primary amines employing Ag$_2$CO$_3$ as catalyst and ($p$-MeOC$_6$H$_4$)$_3$P as an additive. Specifically, they developed a one-pot stepwise strategy to afford oxazolidinones. In this stepwise process, propargylic alcohols react with CO$_2$ to form $\alpha$-alkylidene cyclic carbonates under 2 MPa of CO$_2$. Then, primary amines were directly added without any separation operation. Finally, the oxazolidinones were obtained at 120 °C under atmospheric CO$_2$ pressure (Scheme 9). Moreover, this catalytic system could be reused at least 3 times.

$R_1$ = H, Me, Et
$R_2$ = H, $n$-Bu
$R_1$, $R_2$ = -(CH$_2$)$_5$-
$R_3$ = Me, Et, Ph
$R_4$ = $n$-Bu, $s$-Bu, allyl, Cy, $n$-C$_6$H$_{13}$

**Scheme 7.** AgOAc catalyzed three-component reaction of CO$_2$, propargylic alcohols and primary amines.

$R_1$ = Me
$R_2$ = Me, Ph, Et
$R_1$, $R_2$ = -(CH$_2$)$_5$-
$R_3$ = $n$-Pr, $n$-Bu, Bn, Ph, Cy

**Scheme 8.** AgWO$_4$/Ph$_3$P system catalyzed three-component reaction of CO$_2$, propargylic alcohols and primary amines.

$R_1$ = Me
$R_2$ = Me
$R_1$, $R_2$ = -(CH$_2$)$_5$-
$R_3$ = $n$-Bu, benzyl, Cy

**Scheme 9.** Ag$_2$CO$_3$/($p$-MeOC$_6$H$_4$)$_3$P system catalyzed three-component reaction of CO$_2$, propargylic alcohols and primary amines.

Although transition metal compounds proved to be suitable catalysts for promoting $CO_2$ conversion, most reported systems were costly, unstable, and sensitive to air, water, or light. Therefore, some attention has been drawn to metal-free organocatalysis due to its economy, tunability, and functionality [73,74].

In 1990, Fournier and his colleagues [75] demonstrated that organocatalysts could efficiently promote the three-component tandem reaction of $CO_2$, propargylic alcohols, and primary amines. In the presence of tri-*n*-butylphosphine (Bu$_3$P), 4-methylene oxazolidin-2-ones were generated from the reaction of 2-methyl-3-butyn-2-ol, $CO_2$, and primary amines in 38–72% yields under 5 MPa of $CO_2$ (Scheme 10). Based on this, the Costa group [76] discovered that a series of organic bases and bicyclic guanidines, were a type of favorable catalysts for the reaction of scCO$_2$ with terminal propargylic alcohols and primary amines. The yield of desired products were generally high when primary alkyl-, allyl- and benzylamines were employed, while lower yields were given using primary arylamines (Scheme 11). In 2016, Liu and Hua [77] revealed that both secondary and tertiary propargylic alcohols could efficiently react with primary amines and $CO_2$ under the catalysis of 2,2′,2″-terpyridine in one-pot at 3 MPa of $CO_2$ to afford oxazolidinones and oxazolones (Scheme 12).

**Scheme 10.** Bu$_3$P catalyzed three-component reaction of $CO_2$, propargylic alcohols and primary amines.

**Scheme 11.** MTBD or TBD catalyzed three-component reaction of $CO_2$, propargylic alcohols and primary amines.

**Scheme 12.** 2,2′,2″-terpyridine catalyzed three-component reaction of $CO_2$, propargylic alcohols and primary amines.

ILs are a type of new green organic solvents, which have been applied in numerous areas owing to their unique features such as high stability, nonvolatility, recyclability, and tunability [78,79]. Mainly, ILs are often used as catalysts for the conversion of $CO_2$.

In 2005, Deng et al. [80] demonstrated that even if the metal compounds were absent, the ILs themselves could act as a superior catalyst for this three-component reaction. For example, a range of terminal tertiary propargylic alcohols were able to effectively react with $CO_2$ and primary amines to afford N-substituted 4-methylene-2-oxazolidinones in IL of 1,3-dimethylimidazolium tetrafluoroborate ([DMIm] [$BF_4$]) under 5 MPa of $CO_2$ (Scheme 13).

$R_1$ = Me
$R_2$ = alkyl, Ph
$R_3$ = alkyl benzyl, allyl

**Scheme 13.** [DMIm][$BF_4$] catalyzed three-component reaction of $CO_2$, propargylic alcohols and primary amines.

Sc$CO_2$ has been recognized as an environmentally friendly alternative to organic solvents owing to its nonflammability, easy separation, availability and low cost. In particular, it could be used as a green reaction medium as well as an exceptional raw material [67]. In order to develop an efficient and eco-friendly process for transforming $CO_2$ into oxazolidinones, Xu et al. [81] tried to adopt sc$CO_2$ to react with propargylic alcohols and primary amines in the absence of any additional catalyst and solvent under 14 MPa pressure. Although aniline and tert-butylamine could not produce the desired product, the corresponding 4-methyleneoxazolidin-2-ones were smoothly given in 65–88% yields (Scheme 14).

$R_1$ = Me, Ph, *i*-Pr, *n*-$C_6H_{13}$
$R_2$ = Me
$R_1$, $R_2$ = -(CH$_2$)$_5$-
$R_3$ = *n*-Bu, *s*-Bu, allyl, Cy

**Scheme 14.** Catalyst-free reaction of $CO_2$, propargylic alcohols and primary amines.

Apart from the intensively studied homogeneous catalysis, various kinds of heterogeneous catalysts, including metal–oxides [82], metal nanoparticles [83], molecular sieves [84], silica-supported metal salts [85], and porous polymers [86], etc., were also employed to transform $CO_2$ into high-value chemicals. Recently, metal–organic frameworks (MOFs) have been demonstrated to be favorable solid-state catalysts, owing to their high porosity, thermal stability, structural diversity, and excellent reusability [87,88]. Additionally, MOFs have been regarded as one of the most effective and promising $CO_2$ adsorbents. Therefore, MOFs can significantly heighten the local concentration of $CO_2$ near the catalytic sites to promote the conversion of $CO_2$ [89]. In this aspect, Fei et al. [87] synthesized a Ag(I)-embedded sulfonate-MOF, a non-interpenetrated sulfonate-based porous structure with a prototypical primitive-cubic (pcu) topology, which possessed high $CO_2$ affinity and alkyne activation properties. This MOF could efficiently promote the cyclization reaction of propargylic alcohols with $CO_2$ and the three-component reaction of propargylic alcohols, $CO_2$ and primary amines affording α-alkylidene cyclic carbonates and oxazolidinones with excellent yields under atmospheric pressure (Scheme 15).

Scheme 15. Ag(I)-embedded sulfonate-MOF catalyzed three-component reaction of $CO_2$, propargylic alcohols and primary amines.

Moreover, this Ag(I)-embedded sulfonate-MOF can be reused at least three times. In 2020, Das and Nagaraja [90] reported the one-pot three-component reaction of propargylic alcohols, $CO_2$, and primary amines at room temperature and atmospheric pressure conditions in the presence of a functional MOF-$SO_3$Ag, which was constructed by a Ag(I)-anchored sulfonate-functionalized UiO-66 structure. When 1,8-Diazabicyclo[5.4.0]undec-7-ene (DBU) was employed as an additive, the yield of several oxazolidinones increased to 99%. Further, the recyclability investigation demonstrated that the activity of MOF-$SO_3$Ag would not significantly decrease after five cycles of regeneration, and the original skeleton structure remains intact. In the mechanism study, the high catalytic activity might be ascribed to the synergistic effect between sulfonate functionality and Ag(I) ions, in which Ag(I) activated the C≡C bonds of propargylic alcohols and sulfonates which were responsible for attracting $CO_2$ (Scheme 16).

Scheme 16. (a) The process of the synthesis of MOF-$SO_3$Ag. (b) MOF-$SO_3$Ag catalyzed three-component reaction of $CO_2$, propargylic alcohols and primary amines.

### 2.2. Carbamates as Products

Originally, methods for synthesizing carbamates through a one-pot three-component reaction of $CO_2$, propargylic alcohols, and secondary amines were reported by Bruneau and Dixneuf [91] in 1987, employing $(RuCl_2(Norbornadiene))_n$ as a catalyst in $CH_3CN$. The corresponding carbamates were given in low to moderate yields (Scheme 17). In the same year, an analogous strategy was reported by Sasaki [65], in which $Ru_3(CO)_{12}$ was used to afford carbamates (Scheme 18). In addition to ruthenium compounds, iron and copper complexes were found to be suitable catalysts for this reaction [92,93]. In 1997, Kim et al. [93] demonstrated that a copper complex $[Cu(L)]PF_6$ can effectively promote this three-component reaction under 3.8 MPa of $CO_2$ to afford carbamates. Years later, Jiang et al. [94] reported a AgOAc/DBU system to synthesize β-oxoalkyl carbamates from the reaction of $CO_2$, internal propargylic alcohols and secondary amines in 1,4-dioxane under 2 MPa of $CO_2$. Although the bulky diisopropylamines failed to give the desired products, a range of carbamates could be obtained in 70–93% yields (Scheme 19).

**Scheme 17.** $(RuCl_2(Norbornadiene))_n$ catalyzed three-component reaction of propargylic alcohols, $CO_2$ and secondary amines.

**Scheme 18.** $Ru_3(CO)_{12}$ catalyzed three-component reaction of propargylic alcohols, $CO_2$ and secondary amines.

**Scheme 19.** AgOAc/DBU system catalyzed three-component reaction of propargylic alcohols, $CO_2$ and secondary amines.

Inspired by this case, the synergistic catalytic systems of silver(I) compounds with organic additives have been widely applied in this three-component tandem reaction. In 2014, the He group developed a milder $Ag_2WO_4$/$Ph_3P$ system [71], which was suitable for the three-component reaction of terminal propargylic alcohols, $CO_2$ and secondary amines. Catalyzed by this system, carbamates were obtained in moderate to excellent yields under 0.5 MPa (Scheme 20). Subsequently, the same group found that once $Ag_2WO_4$ was replaced by $Ag_2CO_3$ [95], this three-component tandem reaction could be realized under atmospheric $CO_2$ pressure. The yields of the corresponding β-oxopropylcarbamates were up to 68–98% (Scheme 21a). According to the proposed mechanism, Ag(I) activated the C≡C bonds of propargylic alcohols; meanwhile, the $Ag_2CO_3$ and $Ph_3P$ in situ formed $[(Ph_3P)_2Ag]_2CO_3$ to simultaneously activate the hydroxyls of the propargyl alcohols and $CO_2$, embedding $CO_2$ into propargyl alcohols to generate cyclic carbonate **IV** via intermediates **I**, **II** and **III**. Afterwards, the nitrogen atoms in secondary amines attacked the carbonyl groups of the cyclic carbonate **IV** to produce intermediate **V**, followed by the tautomerization of the enol to the carbamates **3** (Scheme 21b).

Driven by these works, ever increasing attention has been paid to developing the reaction conditions' mildness and economy for the reaction system, especially for the low $CO_2$ pressure and recycling performance of the catalyst. In 2018, Zhang and He et al. [72] reported the $Ag_2CO_3$/$(p-MeOC_6H_4)_3$ system for the synthesis of β-oxopropylcarbamates via a one-pot stepwise method. Similar to the abovementioned synthesis method for oxazolidinones, the first step was to generate α-alkylidene cyclic carbonates under 2 MPa of $CO_2$. Then, the second step involved directly adding secondary amines to synthesize

carbamates under ambient temperature and atmospheric $CO_2$ pressure without any separation operations or additional additives (Scheme 22). Although this strategy required high pressure, the metal loading of this system was extremely low (0.01%), and the catalyst could be recycled at least 3 times. In the same year, a $AgCl/Et_4NCl$ system was proposed by Song, Zhang and Hao et al. [96], which could obtain β-oxopropylcarbamates through a milder and easier strategy. In this reaction, β-oxopropylcarbamates were synthesized in moderate to excellent yields at 60 °C and atmospheric $CO_2$ pressure in $CH_3CN$. Yields of products were mainly affected by the steric hindrance effect and the induction effect of the oxygen atom (Scheme 23).

**Scheme 20.** $Ag_2WO_4/Ph_3P$ system catalyzed three-component reaction of propargylic alcohols, $CO_2$ and secondary amines.

**Scheme 21.** (**a**) $Ag_2CO_3/Ph_3P$ system catalyzed three-component reaction of propargylic alcohols, $CO_2$ and secondary amines. (**b**) The proposed catalytic mechanism of the $Ag_2CO_3/Ph_3P$ system.

**Scheme 22.** $Ag_2CO_3/(p\text{-}MeOC_6H_4)_3$ system catalyzed three-component reaction of propargylic alcohols, $CO_2$ and secondary amines.

**Scheme 23.** $AgCl/Et_4NCl$ system catalyzed three-component reaction of propargylic alcohols, $CO_2$ and secondary amines.

Recently, our group also achieved significant progress in preparing β-oxopropylcarbamates through this one-pot three-component tandem reaction strategy. For example, in 2018, we reported a green and recyclable catalytic system based on AgBr and an IL of 1-ethyl-3-methylimidazolium acetate ($[C_2C_1im][OAc]$) for the three-component tandem reaction of propargylic alcohols, secondary amines, and $CO_2$ under atmospheric $CO_2$ pressure without any solvent (Scheme 24a) [97]. Notably, this $AgBr/[C_2C_1im][OAc]$ system exhibited desirable stability and could be reused at least 5 times. Judging from the yield data, the generation of carbamates was mainly affected by the steric effects of the substituted $R_1$ and $R_2$ groups in the propargylic alcohols. The desired products were obtained in high to excellent yields in the case of most secondary amines. The seldom-reported dissymmetric secondary amine also gave the target product in a moderate yield. Its proposed mechanism is shown in Scheme 24b. Initially, $OAc^-$ and secondary amines simultaneously activated $CO_2$ and the hydroxyl groups of substrate **1** (intermediate **I**) and enhanced the nucleophilicity of the hydroxyl oxygens to the $CO_2$ molecules, leading to the formation of carbonate intermediate **II**. Subsequently, Ag(I) activated the C≡C bond to facilitate the combination of C−O, generating the five-membered ring **III**. Afterwards, the catalysts were released from **III** to form the key intermediate **IV**. Finally, the secondary amine attacked the carbonyl groups of the intermediate **IV** to generate the intermediate **V**, followed by the keto-enol tautomerism and formation of β-oxopropylcarbamates **3**. Moreover, N-heterocyclic carbene (NHC) silver complexes were found in this catalytic system, which might be formed by the interaction between C(2) protons and acetate ions, generating free NHC and then reacting with the Ag salt to create the bis-NHC structure (Scheme 24c).

Apart from the metal-catalyzed process, the metal-free organocatalytic method was also studied. The Costa group [76] studied the reactions of propargylic alcohols with $scCO_2$ or in acetonitrile with gaseous $CO_2$, employing organic bases as catalysts. They found that bicyclic guanidines such as 1,5,7-triazabicyclo[4.4.0]dec-5-ene (TBD) and 7-Methyl-1,5,7-triazabicyclo[4.4.0]dec-5-ene (MTBD) were effective catalysts for transforming $CO_2$ into α-alkylidene cyclic carbonates. Based on this, they successfully obtained a range of carbonates in high yields and good selectivity through a one-step three-component tandem reaction of $CO_2$, terminal propargylic alcohols and an external N-nucleophiles (secondary amines). According to his results, the catalytic performance of MTBD was better than TBD (Scheme 25). In 2007, Qi and Jiang [98] reported that β-oxopropylcarbamates could be efficiently synthesized in the absence of any additional catalysts and organic solvents in

compressed $CO_2$ with propargylic alcohols, secondary amines. The target products were obtained in 35–88% yields at 130 °C and 14 MPa of $CO_2$ (Scheme 26).

**Scheme 24.** (**a**) AgBr/[C$_2$C$_1$im][OAc] system catalyzed three-component reaction of propargylic alcohols, $CO_2$, and secondary amines. (**b**) The proposed catalytic mechanism of the AgOAc/[Emim][OAc] system. (**c**) The formation process of N-heterocyclic carbene (NHC) silver complexes.

Recently, some heterogeneous catalysts were also developed for the target reaction, such as the metal-based nanostructured materials of one-dimensional silver nanowires (Ag NWs) and spherical silver nanoparticles (Ag NPs). Their unique properties (good dispersity and uniformity) and the rapid development of synthesis strategies made it feasible to prepare nanomaterials with different shapes, sizes, structures and tunable compositions [99]. Recently, these kinds of materials were gradually applied in converting $CO_2$ under environmentally friendly conditions. In 2015, the Han group [100] reported the Ag NPs catalyzed cyclic carboxylation reaction of $CO_2$ with propargylic alcohols affording

α-alkylidene cyclic carbonates. Later, Ag NPs combined with MOF was employed to synthesize propionic acid from terminal alkynes and $CO_2$ [54,101].

**Scheme 25.** MTBD or TBD catalyzed three-component reaction of propargylic alcohols, $CO_2$ and secondary amines.

**Scheme 26.** Catalyst-free reaction of propargylic alcohols, $CO_2$ and secondary amines.

Inspired by these works, Qi and Hu et al. [99] successfully synthesized a diversity of β-oxopropylcarbamates via a three-component coupling reaction of propargylic alcohols, $CO_2$ and secondary amines employing Ag NWs as a catalyst for the first time (Scheme 27). In this reaction, terminal propargylic alcohols with various alkyl or aryl substituents could react with symmetric and asymmetric dialkylamines and amines with long alkyl chains to afford target products with good to excellent yields under 1 MPa of $CO_2$ in $CH_3CN$ with DMAP as an additive. Moreover, Ag NWs could be easily recycled by centrifugation and reused at least 4 times. In 2019, Chang and Sadeghzadeh et al. [102] developed ionic gelation (IG) (of TPP and spirulina) related catalyst (DFNS/IG–Ag(I) NPs) for the one-pot synthesis of β-oxopropylcarbamates through the three-component tandem reaction of propargylic alcohols, $CO_2$ and secondary amines at 50 °C and 1.5 MPa of $CO_2$ (Scheme 28). The yields of carbamates were mainly affected by steric hindrance of the substituents in propargylic alcohols or amines. Notably, this catalyst could be recycled 5 times. In the proposed catalytic mechanism, spirulina and TPP synergistically activated the hydroxyl groups of propargyl alcohols. Simultaneously, $CO_2$ was captured and activated by the synergistic effects of TPP and spirulina. Afterwards, Ag(I) species activated the C≡C bonds to facilitate the combination of the negatively charged oxygen atoms with the carbons in the triple bonds, leading to the formation of the five-membered rings. Subsequently, the catalysts were released from the five-membered ring to form the cyclic carbonate intermediate. Finally, the secondary amine attacked the carbonyl groups of the cyclic carbonate intermediates, followed by the keto-enol tautomerism and formation of β-oxopropylcarbamates.

In the same year, Fan et al. [103] developed a unique bifunctional hybrid catalyst (TEMPO–FPS-laccase NPs) through co-immobilization of 2,2,6,6-tetra-methylpiperidine-1oxyl (TEMPO) and laccase in the same cavities into glycidyloxypropyl functionalized fibrous phosphosilicate (FPS) nanoparticles (Scheme 29a). This catalyst was applied for the one-pot synthesis β-oxopropylcarbamates via a three-component tandem reaction of propargylic alcohols, $CO_2$, and secondary amines employing water as the solvent under 1.5 MPa of $CO_2$ (Scheme 29b). The yields of the corresponding carbamates ranged from 48–97%. Although the system required relatively high $CO_2$ pressure, this heterogeneous bio-catalyst was truly attractive due to its environmentally friendly characters such as superior

storage stability, easy separation and remarkable recyclability, which could be reused at least 10 times by extremely low catalyst loading. The remarkable cycle performances were mainly attributed to the FPS nanostructure, while the excellent catalytic performances were determined by the use of enzymes. Similar to the above mechanism, this production of β-oxopropylcarbamates also proceeded with the formation of α- alkylidene cyclic carbonate, nucleophilic attack of amine, and keto-enol tautomerism. In 2020, Hassan et al. [104] reported that in the presence of a new magnetic nano-catalyst, palladium NPs supported on magnetic fibrous silica ionic gelation (FeNi$_3$/DFNS/IG/Pd MNPs), β-oxopropylcarbamates were synthesized through a three-component tandem reaction of propargylic alcohols, $CO_2$ and secondary amines in 54–98% yields at 70 °C and 2 MPa of $CO_2$ with water as a solvent. Moreover, this nano-catalyst can be recycled at least 10 times (Scheme 30).

**Scheme 27.** Ag NWs/DMAP system catalyzed three-component reaction of propargylic alcohols, $CO_2$, and secondary amines.

**Scheme 28.** DFNS/IG–Ag(I) NPs catalyzed three-component reaction of propargylic alcohols, $CO_2$ and secondary amines.

**Scheme 29.** (**a**) The process of co-immobilization of TEMPO laccase and laccase onto glycidyloxypropyl-functionalized FPS nanoparticles; (**b**) TEMPO–FPS-laccase NPs catalyzed three-component reaction of propargylic alcohols, $CO_2$ and secondary amines.

**Scheme 30.** FeNi$_3$/DFNS/IG/Pd MNPs catalyzed three-component reaction of propargylic alcohols, $CO_2$ and secondary amines.

In the past decades, graphene has been widely used as a favorable supporter for metal or metal–oxide nanoparticles due to its large specific surface area [105,106], excellent dispersion for metal particles, and superior stability [107–109]. Based on this, a more gentle strategy was developed by He et al. [109]. They reported the graphene oxide (rGO) supported Ag NPs catalysts (Ag-rGO) for the efficient transformation of $CO_2$ at ambient conditions. Target $\beta$-oxopropylcarbamates could be obtained smoothly from the reaction of 2-methyl-3-butyn-2-ol, diethylamine, and $CO_2$ in 86% yield under atmospheric $CO_2$ pressure in the presence of Ag-rGO and [N$_{4444}$][Triz]. Additionally, the robust stability of Ag-rGO enables its structure and activity to be maintained after 5 cycles (Scheme 31).

**Scheme 31.** Ag-rGO catalyzed three-component reaction of propargylic alcohols, $CO_2$ and secondary amines.

## 3. Three-Component Reactions of Propargylic Alcohols, $CO_2$ and $H_2O$

Since Kutscheroff developed the Hg(II) salts/$H_2SO_4$ catalytic system for the hydration of alkynes [110,111], great efforts have been devoted to a mercury-free route to obtain versatile carbonyl compounds [112–114]. Hydration of propargylic alcohols is a straightforward method to generate $\alpha$-hydroxy ketones, which are vital skeletons in various drugs and natural products [115–117]. However, the direct hydration of propargylic alcohols to prepare $\alpha$-hydroxy ketones was not an ideal route, owing to the formation of byproducts resulting from the Meyer–Schuster and Rupe rearrangements [118,119]. Therefore, a more efficient and eco-friendly reaction route has been developed, employing $CO_2$ as a promoter for hydrating propargylic alcohols and $H_2O$. The procedures of this three-component reaction were involved with the generation of $\alpha$-alkylidene cyclic carbonates, in situ hydrolysis and the keto−enol tautomerization to afford the desired $\alpha$-hydroxy ketones (Scheme 32). In the overall cycle, $CO_2$ was not consumed and acted as a co-catalyst.

**Scheme 32.** The process of $CO_2$-promoted hydration of propargylic alcohols.

Transition-metal compounds such as Ag and Cu salts are favorable catalysts for the $CO_2$-promoted hydration of propargylic alcohols. In 2014, Qi et al. [40] reported that the $CO_2$-promoted process could proceed smoothly in the presence of AgOAc and DBU

under 2 MPa of $CO_2$ in a mixed acetonitrile/water solvent. Both the internal and terminal propargylic alcohols could produce the target products in good yields (Scheme 33). In 2018, Song and Liu et al. [46] discovered that $\alpha$-hydroxy ketones could be efficiently produced in the presence of $ZnCl_2$ and DBU under 1 MPa of $CO_2$ in $CH_3CN$ (Scheme 34). In 2019, Chen et al. [42] developed a $Cu_2O$/DBU system that could efficiently transform a range of propargylic alcohols into desired $\alpha$-hydroxy ketones under atmospheric $CO_2$ pressure in $CH_3CN$ employing cyclohexyldiphenylphosphine as an additive (Scheme 35).

$R_1$ = Me, Et, *i*-Bu
$R_2$ = Me, Et, *i*-Pr, *i*-Bu, *n*-Hex, Ph
$R_1$, $R_2$ = -(CH_2)_4-, -(CH_2)_5-
$R_3$ = H, Me, aryl, thiophene, Py, Ph

**Scheme 33.** AgOAc and DBU catalyzed $CO_2$-promoted hydration of propargylic alcohols.

$R_1$ = Me
$R_2$ = *n*-$C_6H_{13}$, Ph

**Scheme 34.** $ZnCl_2$ and DBU catalyzed $CO_2$-promoted hydration of propargylic alcohols.

$R_1$ = Me
$R_2$ = Et, *i*-Bu, *n*-Hex, vinyl, Ph
$R_1$, $R_2$ = -(CH_2)_5-

**Scheme 35.** $Cu_2O$/DBU system catalyzed $CO_2$-promoted hydration of propargylic alcohols.

In order to synthesize $\alpha$-hydroxy ketones through a greener route, our group developed an effective AgOAc/1-ethyl-3-methylimidazolium acetate ([Emim][OAc]) system for the $CO_2$-promoted hydration of propargylic alcohols to afford $\alpha$-hydroxy ketones under atmospheric $CO_2$ pressure and solvent-free condition (Scheme 36a) [43]. Diverse propargylic alcohols could be transformed into target hydroxy ketones in excellent yields with the catalysis of only a trace amount of silver (0.005−0.25 mol %). Notably, this system could be reused at least 5 times. Furthermore, an unprecedented turnover number (TON) of 9200 was obtained. The proposed catalytic mechanism of the AgOAc/[Emim][OAc] system is shown in Scheme 36b. Firstly, the highly concentrated [OAc] simultaneously activated $CO_2$ and the hydroxyl groups from substrates **1**, inducing the following nucleophilic attack and formation of intermediates **II**. Afterward, Ag species activated the C≡C bonds to promote the formation of the C−O bonds, resulting in the creation of five-membered rings **III**. Then, the catalysts were released and the key $\alpha$-alkylidene cyclic carbonates **IV** were generated. Subsequently, water molecules acted as nucleophiles and attacked the carbonyls of $\alpha$-alkylidene cyclic carbonates **IV** to generate the intermediates **V** with the catalysis of the basic ILs, followed by the keto-enol tautomerism. Finally, one equivalent of $CO_2$ was released and the desired ketone **2** was formed.

**Scheme 36.** (**a**) AgOAc/[Emim][OAc] system catalyzed $CO_2$-promoted hydration of propargylic alcohols; (**b**) The proposed catalytic mechanism of the AgOAc/[Emim][OAc] system.

In 2015, Liu et al. [41] reported a metal-free catalytic system for hydrating propargylic alcohols and $H_2O$ with $CO_2$ as a co-catalyst to produce α-hydroxy ketones at atmospheric $CO_2$ pressure (Scheme 37). A series of desired products could be obtained in good to excellent yields in the presence of tetrabutylphosphonium imidazole ([Bu$_4$P][Im]). More investigations revealed that both $CO_2$ and [Bu$_4$P][Im] were indispensable for this reaction. Additionally, [Bu$_4$P][Im] acts as the catalyst and solvent, which could be easily separated and reused at least 5 times without obvious activity loss.

**Scheme 37.** [Bu$_4$P][Im] catalyzed $CO_2$-promoted hydration of propargylic alcohols.

## 4. Three-Component Reactions of Propargylic Alcohols, $CO_2$ and Monohydric Alcohols

As weak nucleophiles, monohydric alcohols could also react with propargylic alcohols and $CO_2$, affording β-oxoalkyl carbonates, which are a kind of dissymmetric carbonate and generally used as important reagents and intermediates in organic synthesis [120,121]. The three-component reaction of propargylic alcohols, $CO_2$ and monohydric alcohols could

proceed smoothly through the sequential steps of carboxylative cyclization, nucleophilic attack of monohydric alcohols, and tautomerization (Scheme 38).

**Scheme 38.** The process of the three-component reaction of propargylic alcohols, $CO_2$ and monohydric alcohols.

Dixneuf et al. [122] initially first synthesized cyclic carbonates using tributylphosphine as a catalyst. Then these cyclic carbonates react with monohydric alcohols to afford desired dissymmetric carbonates in the presence of triethylamine or KCN (Scheme 39). Later, the Costa group [76] developed the one-pot synthesis strategy to obtain β-oxoalkyl carbonates through the three-component reaction of propargylic alcohols, $CO_2$ and monohydric alcohols employing bicyclic guanidines as catalysts under $scCO_2$ conditions.

**Scheme 39.** Two step strategy for the three-component reaction of propargylic alcohols, $CO_2$ and monohydric alcohols.

In recent years, silver and zinc compounds have been developed as excellent catalysts for this three-component reaction. In 2016, He et al. [44] reported an effective $Ag_2CO_3/Ph_3P$ system for the target reaction under 1 MPa of $CO_2$ in $CH_3CN$ (Scheme 40). In 2017, Ma and Han et al. [123] developed a AgCl/1-butyl-3-methylimidazolium acetate ([Bmim][OAc]) system, which could catalyze this three-component reaction under ambient conditions without any solvents. In addition, this catalytic system could be recycled 5 times (Scheme 41). In 2018, Song and Liu et al. [46] explored non-noble $ZnCl_2$ and DBU as the catalytic system for the synthesis of β-oxopropyl carbonates under 1 MPa of $CO_2$ in $CH_3CN$ (Scheme 42).

$R_1$ = H, alkyl
$R_2$ = alkyl
$R_3$ = alkyl, aryl, Py, thiophene

**Scheme 40.** $Ag_2CO_3/Ph_3P$ system catalyzed three-component reaction of propargylic alcohols, $CO_2$ and monohydric alcohols.

Although the above strategies could effectively synthesize β-oxoalkyl carbonates, several drawbacks remain, such as high $CO_2$ pressure, narrow substrate scope, etc. To improve the reaction conditions, Song and Zhang et al. [47] developed a synergistic silver sulfadiazine/$^nBu_4NBr$ system for the one-pot synthesis of β-oxopropyl carbonates under atmospheric $CO_2$ pressure and solvent-free conditions (Scheme 43). Additionally, this protocol showed excellent tolerance for a wide range of propargyl alcohols and monohydric alcohols. Various desired carbonates could be obtained in 64–99% yields, although phenol failed to afford the target product, and an excess amount of alcohol was generally required. Furthermore, they speculated that the excellent catalytic performance was attributed to the

synergistic catalysis of silver sulfadiazine and $^{n}Bu_4NBr$. In 2020, Detrembleur et al. [51] proposed a AgI/tetrabutylammonium phenolate ([TBA][OPh]) catalytic system, which was highly active for this one-pot process (Scheme 44). Most of the desired products could be obtained in more than 97% yields under atmospheric $CO_2$ pressure in DMSO. The yield data further demonstrated that primary monohydric alcohols were more active than secondary or tertiary monohydric alcohols. Notably, the authors explored the kinetic insights and the catalytic mechanism through FT-IR spectroscopy and density functional theory (DFT) calculations.

$R_1$ = Me
$R_2$ = Me, Et, *i*-Bu
$R_1,R_2$ = -$(CH_2)_5$-
$R_3$ = Me, Et, *n*-Bu, benzyl, furan

**Scheme 41.** AgCl/[Bmim][OAc] system catalyzed three-component reaction of propargylic alcohols, $CO_2$ and monohydric alcohols.

**Scheme 42.** $ZnCl_2$/DBU system catalyzed three-component reaction of propargylic alcohols, $CO_2$ and monohydric alcohols.

$R_1$ = H, Me
$R_2$ = Me, Et, *i*-Bu, *n*-$C_5H_{11}$, vinyl
$R_1, R_2$ = -$(CH_2)_5$-
$R_3$ = aryl, *n*-Oct, *n*-Pr, Cy, Ph

**Scheme 43.** Silver sulfadiazine/$^{n}Bu_4NBr$ system catalyzed three-component reaction of propargylic alcohols, $CO_2$ and monohydric alcohols.

$R_1$ = Me
$R_2$ = Me, Et, *i*-Bu
$R_3$ = *n*-Bu, *i*-Pr, benzyl

**Scheme 44.** AgI/[TBA][OPh] catalyzed three-component reaction of propargylic alcohols, $CO_2$ and monohydric alcohols.

Recently, natural minerals have also been designed as catalysts for the transformation of $CO_2$. In 2020, Liu et al. [50] loaded silver on attapulgite (ATP) material, a hydrous silicate clay mineral-rich in Mg and Al with a fibrous morphology and demonstrated a certain adsorption effect on $CO_2$. This effective Ag/ATP nanocomposite catalyst was applied for

the one-pot synthesis of non-symmetric carbonates through the three-component reaction of propargylic alcohols, $CO_2$ and monohydric alcohols (Scheme 45a). The desired β-oxoalkyl carbonates were obtained in low to high yields under 1 MPa of $CO_2$ in DMF with 1,5-diazabicyclo[4.3.0]non-5-ene (DBN) as a co-catalyst. Moreover, the yields were largely influenced by the electron-donating effect of $R_3$ in alcohols. In the investigations on recyclability, the authors found that Ag/ATP could be recycled at least 10 times by increasing the reaction temperature to decompose the produced carbonates coated on the catalyst's surface. Regarding the catalytic mechanism, the hydroxyl protons in propargyl alcohols firstly interacted with DBN to form $[DBNH]^+$. Then the adsorbed $CO_2$ on the ATP was attacked by the hydroxyl and oxygen anions to form intermediates **II**. Subsequently, the silver nanoparticles on the ATP activated the C≡C to promote the connections of the oxygen atoms from the $CO_2$ to the carbon atoms in the triple bonds, resulting in the formation of five-membered rings **III**. Afterward, intermediates **III** received the protons from $[HDBN]^+$ and gave the key α-alkylidene cyclic carbonates **IV**. Eventually, DBN activated the hydroxyl groups in monohydric alcohols to attack the carbonyl groups of the cyclic carbonates **IV** to produce intermediates **V**. The target β-oxoalkyl carbonates **3** were finally formed by the following tautomerism (Scheme 45b).

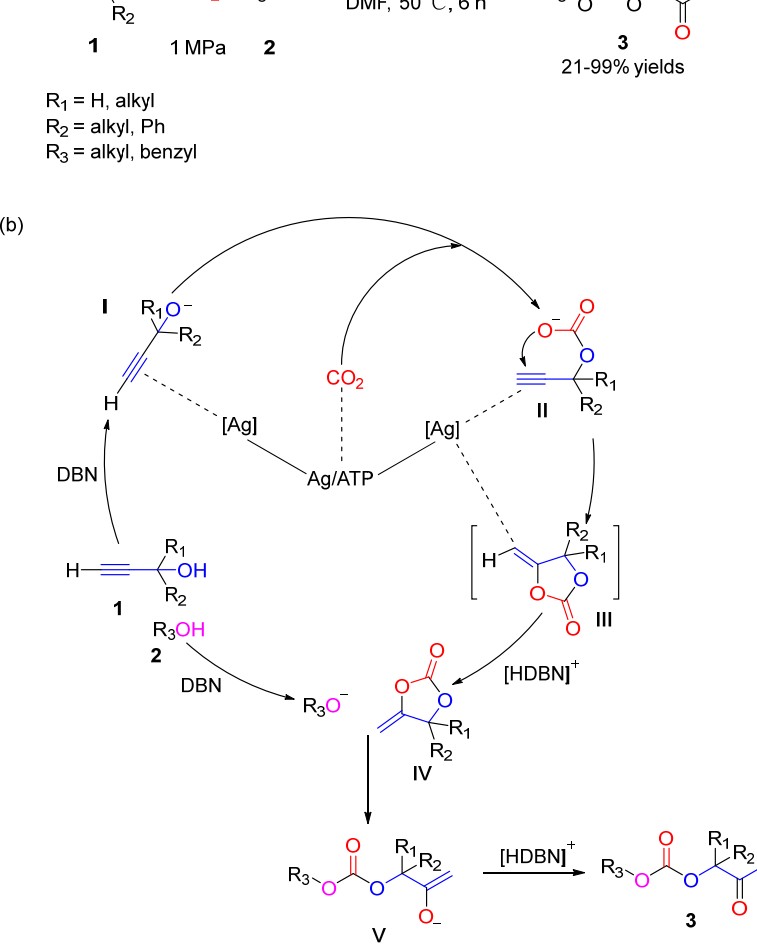

**Scheme 45.** (**a**) Ag/ATP nanocomposite and DBN system catalyzed three-component reaction of propargylic alcohols, $CO_2$ and monohydric alcohols. (**b**) The proposed catalytic mechanism of the Ag/ATP nanocomposite and DBN system.

## 5. Three-Component Reactions of Propargylic Alcohols, $CO_2$ and Bi-Nucleophiles

In the area of CCU, the condensation of vicinal diols or 2-aminoethanols with $CO_2$ has attracted significant attention. Many strategies have been reported for this reaction to afford the corresponding cyclic carbonates or 2-oxazolidinones, which are a series of high-value chemical intermediates [124–130]. Unfortunately, these condensations are usually incomplete due to the thermodynamic limitations and invalidation of catalysts caused by the formation of by-products such as water. Although dehydrating agents are typically applied to overcome the equilibrium limitation [126,127,130], the other by-products derived from the consumed additives still remained in this process [58]. As a result, the yields of desired products in these reactions were usually unsatisfactory even under harsh reaction conditions, which largely limits its practical application. Delightedly, a significant breakthrough has been achieved recently. Scientists revealed that introducing propargylic alcohols into these reactions could effectively avoid the formation of water and bypass the thermodynamic limitation. In this method, the reverse reactions were completely suppressed due to the low nucleophilicity of the generated $\alpha$-hydroxy ketone [45,49,53]. More importantly, these three-component reactions of propargylic alcohols, $CO_2$ and vicinal diols or 2-aminoethanols could simultaneously afford the high-value products of cyclic carbonates or 2-oxazolidinones. Regarding the mechanism, this thermodynamically favorable process undertook a two-step procedure: (1) the formation of $\alpha$-alkylidene cyclic carbonate intermediates **M1** from the carboxylative cyclization of propargylic alcohols and $CO_2$; (2) the nucleophilic ring-opening reaction of intermediates **M1** and vicinal diols or 2-aminoethanols to produce **M2** or **M3**, followed by the generation of cyclic carbonates **2** or 2-oxazolidinones **3** with 1 equiv. of $\alpha$-hydroxyl ketones through intramolecular nucleophilic cyclization (Scheme 46).

**Scheme 46.** The process of three-component reaction of vicinal diols or 2-aminoethanols with propargylic alcohols and $CO_2$.

### 5.1. Vicinal Diols as Bi-Nucleophiles

In the past few years, considerable progress on metal-catalyzed systems for this reaction has been achieved. In 2017, He et al. [45] first proposed a Ag(I)-catalyzed system for this three-component reaction of propargylic alcohols, $CO_2$ and vicinal diols to afford cyclic carbonates and $\alpha$-hydroxyl ketones with an electron-rich bidentate phosphine 4,5-Bis(diphenylphosphino)-9,9-dimethylxanthene (Xantphos) as an additive (Scheme 47). They proved this process was a thermodynamically favorable route through DFT calculations, in which desired products could be obtained in good to excellent yields under 1 MPa of $CO_2$ in $CH_3CN$ catalyzed by the $Ag_2CO_3$/Xantphos system. Subsequently, Song and Liu et al. [46] developed a Zn(II)-catalyzed system for this three-component reaction (Scheme 48) in 2018. Target products could be obtained in excellent yields under 1 or 2 MPa in $CH_3CN$. However, an excess of vicinal diols (1.5 equiv.) was required. Moreover, both of these routes required solvents and high $CO_2$ pressure.

**Scheme 47.** (**a**) Ag$_2$CO$_3$/Xantphos system catalyzed three-component reaction of propargylic alcohols, CO$_2$ and vicinal diols. (**b**) The structure of Xantphos.

**Scheme 48.** ZnCl$_2$/DBU system catalyzed three-component reactions of propargylic alcohols, CO$_2$ and vicinal diols.

In 2019, Song and Zhang et al. [48] developed an efficient silver sulfadiazine/Et$_4$NBr synergistic catalytic system, enabling this three-component reaction to proceed smoothly under atmospheric CO$_2$ pressure and solvent-free conditions (Scheme 49a). Furthermore, this catalytic system exhibited a broad substrate scope. Propargylic alcohols with different alkyl substituents, including long-chain alkyl, isopropyl, cyclohexyl, vinyl, and phenyl, could react smoothly with 1,2-diols bearing short- or long-chain alkyl and sterically hindered substituents. The corresponding cyclic carbonates and $\alpha$-hydroxyl ketones were obtained in satisfactory yields under optimal conditions. Moreover, a plausible mechanism was speculated, as shown in Scheme 49b. Initially, sulfadiazine and Ag species simultaneously activated $-$OH and C≡C of the propargylic alcohols to form the carbonate intermediates **II**. Significantly, Et$_4$N$^+$ stabilized the carbonate intermediates **II** and enhanced the nucleophilicity of the oxygen atoms. Then, the catalysts were released and the cyclic carbonates **III** were generated. Afterward, the activated $-$OH from vicinal diols attacked the carbonyl groups of intermediates **III** to produce intermediates **IV**. Eventually, the target cyclic carbonates **3** were produced through the intramolecular nucleophilic cyclization with 1 equiv. of $\alpha$-hydroxyl ketones **4** released from the skeletons.

Regarding organocatalysis, strong bases such as DBU, TBD, and MTBD, etc., are beneficial for the activation of CO$_2$ and hydroxyl groups due to their rich electronegativity [52]. Based on this, Song and Liu et al. [52] reported a facile metal-free DBU-catalyzed strategy for the one-pot preparation of cyclic carbonates and $\alpha$-hydroxy ketones from propargylic alcohols, CO$_2$ and vicinal diols. However, the reaction required a large amount of solvent (DMF) and relatively high CO$_2$ pressure (3 MPa) (Scheme 50). Subsequently, mild N-heterocyclic olefins–CO$_2$ adducts (NHO–CO$_2$)/MTBD organic catalytic system was reported by Zhou and Lu et al. [49]. In the presence of this system, various functionalized five-membered cyclic carbonates could be obtained under ambient conditions (Scheme 51a). Notably, this system showed high tolerance for a wide range of vicinal diols. The corresponding cyclic

carbonates could be produced in moderate to high yields. Specifically, this three-component reaction proceeded via a one-pot two steps strategy. The initial carboxylative cyclization was catalyzed by the NHO-$CO_2$ adducts under ambient conditions. Once propargylic alcohols were completely converted, MTBD, vicinal diols and solvents were added successively for the subsequent transesterification. Additionally, the authors employed complicated polyhydroxy carbohydrate derivatives for this one-pot method to accurately synthesize bio-based cyclic carbonates.

**Scheme 49.** (**a**) Silver sulfadiazine/$Et_4NBr$ synergistic catalytic system catalyzed three-component reaction of propargylic alcohols, $CO_2$ and vicinal diols. (**b**) The proposed catalytic mechanism of silver sulfadiazine/$Et_4NBr$ synergistic catalytic system.

**Scheme 50.** DBU catalyzed three-component reaction of propargylic alcohols, $CO_2$ and vicinal diols.

**Scheme 51.** (**a**) NHO–$CO_2$/MTBD organic catalytic system catalyzed three-component reaction of propargylic alcohols, $CO_2$ and vicinal diols. (**b**) The structure of NHO–$CO_2$ and MTBD.

## 5.2. Aminoethanols as Bi-Nucleophiles

Recently, Ag(I) and Cu(I) compounds have been found to be efficient catalysts for this three-component reaction. In 2016, the He group [53] first reported the $Ag_2CO_3$/Xantphos system for the three-component reaction of propargylic alcohols, $CO_2$, and 2-aminoethanols, which could smoothly convert diverse substrates into the corresponding 2-oxazolidinones and a-hydroxyl ketones under 1 MPa of $CO_2$ in $CHCl_3$ (Scheme 52). This reaction proceeded through the sequential steps of carboxylative cyclization, ring-opening and intramolecular nucleophilic cyclization. Subsequently, they reported a similar $Ag_2O$/1,1,3,3-tetramethylguanidine (TMG) system for this three-component reaction, which could proceed successfully under 1 MPa of $CO_2$ in $CH_3CN$ with a TON up to 1260 (Scheme 53) [54]. In addition to the Ag(I) catalytic systems, more economical Cu(I) catalytic systems were developed for this three-component reaction by the same group in 2018. The desired products could be obtained in good to excellent yield in the presence of CuI, 1,10-phen, and *t*-BuOK under 0.5 MPa of $CO_2$ (Scheme 54) [55]. Although significant progress has been achieved, these catalytic systems still suffered from several shortcomings such as the requirement for high $CO_2$ pressure, the addition of solvents or additives, low catalyst recyclability, and high metal loading, which limited their further applications in industry.

**Scheme 52.** $Ag_2CO_3$/Xantphos system catalyzed three-component reaction of propargylic alcohols, $CO_2$, and 2-aminoethanols.

**Scheme 53.** $Ag_2O$/TMG system catalyzed three-component reaction of propargylic alcohols, $CO_2$, and 2-aminoethanols.

**Scheme 54.** CuI, 1,10-phen and *t*-BuOK catalyzed three-component reaction of propargylic alcohols, $CO_2$, and 2-aminoethanols.

Inspired by these works, our group combined the advantages of transition-metal catalysts and ILs and developed a favorable $AgNO_3$/$[C_2C_1im][OAc]$ system for this reaction (Scheme 55a) [57]. A wide range of 2-oxazolidinones and $\alpha$-hydroxyl ketones could be simultaneously obtained in excellent yields under atmospheric $CO_2$ pressure at 60 °C without any additives or traditional volatile solvents with an extremely low metal loading (0.25% $AgNO_3$). In addition, this system showed excellent recyclability, which could be easily recycled and reused 5 times. Moreover, this system exhibited outstanding performance in evaluating the green metrics. In the aspect of the catalytic mechanism, the basic $OAc^-$ firstly activated the $-OH$ in substrates **2**, promoting its interaction with $CO_2$ to form the carbonate intermediates **II**. Next, the Ag species activated the $C\equiv C$ bonds, resulting in the combination of the negatively charged oxygen atoms with the carbon atoms in the triple bonds to afford the five-membered rings **III**. Subsequently, the catalysts were released from the five-membered rings and intermediates **IV** were generated. Afterwards, intermediates **V** were produced through the nucleophilic attack of the carbonyl groups in intermediates **IV** by the nitrogen atoms of the substrates **2**. Then the keto-enol tautomerism occurred, leading to the formation of intermediates **VI**. Finally, the desired 2-oxazolidinones **3** were generated via the intramolecular nucleophilic cyclization of intermediates **VI** with $\alpha$-hydroxyl ketones **4** released from the molecules (Scheme 55b). Later, we demonstrated that the cheaper Cu(I) salt could also be employed as an effective catalyst for this reaction when combined with ILs (Scheme 56) [58]. In the presence of CuBr and 1-butyl-3-methylimidazolium acetate ($[C_4C_1im][OAc]$), a variety of desired products could be produced in good to excellent yields under 1 atm of $CO_2$ pressure with a low metal loading (0.5 mol% of CuBr). This system also did not require additional volatile organic solvents and additives. Moreover, it could be reused at least 3 times.

**Scheme 55.** (**a**) $AgNO_3/[C_2C_1im][OAc]$ system catalyzed three-component reaction of propargylic alcohols, $CO_2$, and 2-aminoethanols. (**b**) The proposed catalytic mechanism of $AgNO_3/[C_2C_1im][OAc]$ system.

**Scheme 56.** $CuBr/[C_4C_1im][OAc]$ system catalyzed three-component reaction of propargylic alcohols, $CO_2$, and 2-aminoethanols.

In 2018, Li and He et al. [56] synthesized a task-specific IL, 1,5,7-triazabicylo[4.4.0]dec-5-ene trifluoroethanol ([TBDH][TFE]) through an anion-exchange resin, for the green synthesis of 2-oxazolidinones and α-hydroxyl ketones based on the three-component reaction of propargylic alcohols, $CO_2$ and 2-aminoethanols under atmospheric $CO_2$ pressure at 80 °C (Scheme 57). These ILs act as both catalysts and solvents, which could be easily recycled and reused at least for 5 times without significant loss of activity.

**Scheme 57.** [TBDH][TFE] catalyzed three-component reaction of propargylic alcohols, $CO_2$, and 2-aminoethanols.

## 6. Isomerization of $\alpha$-Alkylidene Cyclic Carbonates

Recently, the isomerization of $\alpha$-alkylidene cyclic carbonates generated from the carboxylative cycloaddition of $CO_2$ into propargylic alcohols was developed to afford valuable tetronic acids, which are a series of crucial intermediates and essential skeletons in agricultural, pharmaceutical, and biological chemistry [131–133]. Compared with the already reported route [59,134–139], this route showed the advantages of easily accessible raw materials, mild reaction conditions, simple operation, high atom economy, etc. Consequently, it is an environmentally friendly alternative strategy, which provides a new methodology for the subsequent derivative applications of $\alpha$-alkylidene cyclic carbonates.

AnUntil now, the only example for synthesizing tetronic acids through the reactions of propargylic alcohols and $CO_2$ was reported in 2018. Zhou and Yu et al. [59] demonstrated that a wide range of tetronic acid derivatives could be obtained in 58–90% yields through the tandem reactions of cyclization and isomerization in the presence of excessive $Cs_2CO_3$ at atmospheric $CO_2$ pressure and 65 °C in 1,3-dimethyl-2-imidazolidinone (DMI) (Scheme 58a). This system showed excellent tolerance for functional groups. The substrates with both symmetric and asymmetric dialkyl, methyl, and phenyl at $\alpha$-position of the hydroxyl group, and cyclic, tertiary alcohols with various halogen substituents could undergo this process smoothly and obtain the target products with satisfactory yields. Based on the yield data, the aryl propargylic alcohols with electron-withdrawing groups were more favorable than those with electron-donating groups. The proposed mechanism for this reaction is shown in Scheme 58b. First, the $CO_2$ reacts with the proton-removed substrates **1** to form carbonate intermediates **I**. Then the $\alpha$-alkylidene cyclic carbonate intermediates **II** are formed through the intramolecular cyclization of intermediates **I**. Afterwards, the carbonate anions attacked the carbonyl groups in intermediates **II** and opened the rings to form intermediates **III**. Next, the carbonate anions were released via Dieckmann-type condensation and the intermediates **IV** were formed. Finally, the intermediates **IV** underwent further deprotonation by addition of excess $Cs_2CO_3$, resulting in the formation of enolates **V**, which could be quenched by the acids to produce the desired tetronic acids **2**.

**Scheme 58.** (a) $Cs_2CO_3$-catalyzed reaction of propargylic alcohols with $CO_2$. (b) The proposed catalytic mechanism of $Cs_2CO_3$.

## 7. Conclusions

In summary, the tandem reactions based on the cyclization of $CO_2$ and propargylic alcohols were summarized in this review. Notably, the three-component tandem reaction of $CO_2$, propargylic alcohols with nucleophiles and the isomerization of $\alpha$-alkylidene cyclic carbonates could afford a series of vital skeletons in organic synthesis, biology, and pharmacy. Moreover, these tandem routes exhibited great economic and environmental benefits. Currently, catalytic systems have been increasingly developed for these reactions, aiming to remove volatile organic solvents, and improve catalytic activity and recyclability. However, the reported reaction systems still have some drawbacks such as dependence on high catalyst equivalent and high purity of $CO_2$, which limits their large-scale industrial application. Therefore, the development of catalysts with low catalyst loading to reduce costs and a higher $CO_2$ adsorption to utilize low-concentration $CO_2$ are urgently required. These achievements will expand this area into a more important branch of the future CCU strategy.

**Author Contributions:** Conceptualization, Y.Y.; methodology, B.J. and X.Y.; writing—original draft preparation, X.Y., Y.G. and Y.X.; writing—review and editing, B.J., X.Y., Y.G., N.L., H.D.V. and Y.Y.; supervision, Y.Y. and F.V.; funding acquisition, F.V. and Y.G. All authors have read and agreed to the published version of the manuscript.

**Funding:** This research was funded by the National Natural Science Foundation of China (No. 22102127) and the Foundation (No. GZKF202023) of State Key Laboratory of Biobased Material and Green Papermaking, Qilu University of Technology, Shandong Academy of Sciences.

**Data Availability Statement:** All data were taken from the articles of the bibliography section.

**Conflicts of Interest:** The authors declare no conflict of interest.

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
