# Peer review of "Tandem Reactions Based on the Cyclization of Carbon Dioxide and Propargylic Alcohols: Derivative Applications of α-Alkylidene Carbonates"

_catalysts, doi:10.3390/catal12010073_

Round 1
Reviewer 1 Report
This review summarized the development and the frontier of the related fields of efficient CO2 conversion via three-component tandem reactions, including the favorable paths of CO2 conversion, the efficient catalytic systems and the reaction mechanism in the form of a combination of pictures and text. The research ideas and development trends were fully pointed out. Moreover, it established an integral research framework system by reviewing the relevant literature, and provided a set of feasible green conversion ideas for the utilization of CO2. The paper is well written, logical and informative. Consequently, I recommended this manuscript to be published in Catalysts. However, this manuscript would be more complete after the minor modification following the suggestions below:
- At the end of the second paragraph of the introduction, it is mentioned that " new synthetic routes and high-efficiency catalysts" And the third paragraph summarizes the synthetic routes, but did not mention high-efficiency catalysts at all. The context is not close and needs to be improved.
- The full text made a systematic summary of carbon dioxide through the three-component tandem reaction, but did not summarize the current deficiencies of the research in this field. The deficiencies should be put forward, and the improved viewpoints and novel research ideas should be given in section conclusion.
- Some paragraphs are too long, such as the paragraph starting from line 107, 273, and 663.
- There are some formatting, typo or grammatical errors in the manuscript, which were listed as follows:
(1) Line 119: "Based In 2007" should be changed to "In 2007" or "Based on this, in 2007".
(2) Line 673, 677 and 682 "Moreover". The same progressive word appears three times in a row, so the Line 677 "Moreover" should be changed to "In addition/Furthermore"
Reviewer 2 Report
General comment: The review is written well-structured with up-to-date information of literature. It explains the role and usage of carbon dioxide in tandem reactions based on the cyclization of CO2 and propargylic alcohols were summarized. Also, the authors discuss a number of ways of CO2 utilizations, including hydration, amination, isomerization of hydroxyl ketones, oxazolidinones, carbamates, and natural carboxylic acid and their carbonates. Finally, the authors conclude the environmental benefits of avoiding unwanted catalysts. Still, there are some grammatical and typical errors that need to fix it before publishing the review.
- In the abstract, the following grammatical errors should correct it
Line 19. ….. “have attracted” should be “has attracted”
Line 20. ….. “attentions” should change as “attention”
Line 21. …… “nonflammable and renewable” as “nonflammable, and renewable”
Line 21. …… “transformation” as “the transformation”
Line 25. ……”are a series” as “which are a series” and “which supplied” as “that supplied”
Line 27. …… “attentions have” should be “attention has”
- Line 52. The authors written CO2 is a non-toxic gas. As per the literature reports, CO2 also has a poisoning nature at higher concentrations. We can’t say it is an exactly non-toxic gas. So, better to rewrite about its toxicity or to remove.
- Figure 1 is not linked in the text.
- In Figure 1, instead of the Ball-still model of CO2 structure, Chemdraw structure will be a clear vision for readers.
- Regarding the correlation between the presented scheme number and cited scheme number in the text is not matching in many places. Eg. Scheme 24. Running text discussed Scheme 24a, 24b, 24c….. etc. But unable to find like that in this review.
- Page 29 deals, Scheme 57, and running text denote Scheme 57b. But there is no Scheme 57 in the text and there is no Scheme 57b in the Scheme.
- Scheme 58 is not linked with running text.
- Authors should check the whole manuscript of the connectivity between the captions and the captions in the running text. Otherwise, the present version may make confuse the researcher while reading the review.
